# Helium Abundance Decrease in ICMEs in 23–24 Solar Cycles

**Alexander A. Khokhlachev \*, Yuri I. Yermolaev, Irina G. Lodkina, Maria O. Riazantseva**
**and Liudmila S. Rakhmanova**

Space Research Institute, Russian Academy of Sciences, 117997 Moscow, Russia
\*  Correspondence: aleks.xaa@yandex.ru

**Abstract:** Based on the OMNI database, the influence of the solar activity decrease in solar cycles (SCs) 23–24 on the behavior of the relative helium ions abundance $N_\alpha/N_P$ inside interplanetary coronal mass ejections (ICMEs) is investigated. The dependences of the helium abundance on the plasma and interplanetary magnetic field parameters in the epoch of high solar activity (SCs 21–22) and the epoch of low activity (SCs 23–24) are compared. It is shown that $N_\alpha/N_P$ significantly decreased in SCs 23–24 compared to SCs 21–22. The general trends of the dependences have not changed with the change of epoch, but the helium abundance dependences on some parameters (for example, the magnitude of the interplanetary magnetic field) have become weaker in the epoch of low activity than they were in the epoch of high activity. In addition, the dependence of the helium abundance on the distance from spacecraft to the ICME axis was revealed; the clearest dependence is observed in magnetic clouds. The $N_\alpha/N_P$ maximum is measured at the minimum distance, which confirms the hypothesis of the existence of a helium-enriched electric current inside an ICME.

**Keywords:** solar wind; interplanetary coronal mass ejection; magnetic cloud; helium abundance

## 1. Introduction

One of the most important characteristics of the solar wind (SW) is the relative abundance of minor ion components; in particular, the most abundant of them, doubly ionized helium ions (alpha particles), $N_\alpha/N_P$ [1–7]. SW is a non-stationary plasma flow in which there are different variations of physical properties at several spatial scales [8–10]. On the one hand, various kinds of plasma processes and instabilities are manifested at small scales ($<10^5$ km, where local processes can make a significant contribution) and medium scales $10^5$–$10^6$ km of magnetohydrodynamic structures (for example, Alfven waves or flux tubes), which can lead to local features of the ion composition [11–14]. On the other hand, large-scale structures with a scale more than $10^6$ km, which are also called "streams" or "phenomena" in the scientific literature, do not have enough time to change the ion composition when moving from the solar corona to the Earth's orbit, and allow one to explore the regions of their origin [15,16]. The classification and detailed description of various large-scale SW phenomena are discussed in a number of papers (see, for example, the papers [17,18] and references therein).

Interplanetary coronal mass ejections (ICMEs) are one of the large-scale SW phenomena. According to spacecraft measurements, the average duration of ICME in the Earth's orbit is about 30 h [17,19]. Their spatial scale along the Sun–Earth axis at an average speed of ~400 km/s is about $10^7$ km. Compared with slow undisturbed SW streams, these phenomena are characterized by an increased relative helium abundance [5,20,21]; however, a high helium abundance itself is not an unambiguous criterion of ICMEs [22]. ICMEs include two types of streams: magnetic clouds (MCs) and Ejecta. They differ in the magnitude and time profile nature of the interplanetary magnetic field (IMF): MCs have a higher and smoothly changing magnetic field. The differences between MCs and

Ejecta are described in more detail in the paper [23].

Similar to many other SW parameters (for example, temperature or IMF magnitude), the relative helium abundance depends on the current activity of the Sun and can vary significantly depending on the phase of the solar cycle (SC) [15,24]. In the papers [25,26], changes in the values of plasma and magnetic field parameters were considered in the period from 1976 to 2019, i.e., solar cycles from 21 to 24. When analyzing the data, selection was carried out both by SC phases and by types of large-scale SW phenomena using the corresponding catalog (http://www.iki.rssi.ru/pub/omni [17], accessed on 19 April 2022). The studies showed that in the solar minimum phase between SCs 22 and 23, there is a significant decrease in the values of plasma and IMF parameters in all types of SW and in all SC phases. These values remain reduced at SCs 23–24; for example, the helium abundance decreased by ~30% compared to SCs 21–22. In one of our recent papers [19], we also showed that the time profiles of all SW parameters in all types of large-scale phenomena in SCs 23–24 have a similar shape with the same profiles in SCs 21–22, but located at lower values. In addition, there is a sharp drop in the number of ICMEs in the last two cycles [25,27]. These results indicate a weakening of the solar wind during the Sun's transition from the epoch of high activity in SCs 21–22 to the epoch of low activity in SCs 23–24. These epochs characterize the overall strength of the respective cycles: the sunspot number, which is a solar activity proxy, was higher in SCs 21–22 than in the last two cycles (see, for example, the paper [25] for more details).

In our recent research [28], the relationship between the relative helium abundance and several SW parameters inside ICMEs during SCs 21–24 was investigated based on statistical analysis. The results of the analysis showed that the helium abundance in an ICME correlates with the IMF magnitude and, as a consequence, the helium abundance increases with a decrease in the proton parameter $\beta$, which is the ratio of the thermal proton pressure to the magnetic one. Such a relationship between the helium abundance and the parameter $\beta$ in an ICME was previously revealed when comparing the time profiles of parameters calculated for different SW streams by the double superposed epoch method [22,29]. This fact was considered as an indication that a helium-enriched electric current exists inside ICMEs. However, the aforementioned paper did not compare the behavior of helium ions in different epochs of solar activity.

In this paper, the dependences of the helium abundance on parameters inside ICMEs are considered in two periods: in the epoch of high solar activity from 1976 to 1996 (SCs 21–22), and in the epoch of low activity from 1997 to 2019 (SCs 23–24). The main goal of the research is to find out how dependences in MCs and Ejecta have changed after the change of solar activity epoch at the end of the 20th century.

## 2. Data and Methods

The study uses the same data sources for the period from 1976 to 2019 as in the papers [19,25,28]. Firstly, hourly average values of SW parameters from the OMNI database were obtained (http://omniweb.gsfc.nasa.gov [30], accessed on 1 February 2022). Further in the text, these values, which are averaged over hourly intervals, will be referred to as "measurement points". Secondly, information about ICME time intervals (divided into MCs and Ejecta) was obtained from the catalog of large-scale SW phenomena (http://www.iki.rssi.ru/pub/omni [17], accessed on 19 April 2022). The number of MCs and Ejecta for the above period is 202 and 1461, respectively. Similar to the paper [28], intervals with incomplete data and measurement points falling on ICMEs boundaries were not taken into account in the subsequent analysis.

For two types of ICME phenomena and for two epochs of high and low solar activity (four cases in total), we consider the relationship of the helium abundance $N_\alpha/N_P$ with the IMF magnitude B, the density $N_P$ and temperature of protons T, and the calculated parameters: thermal pressure $N_P kT$, plasma parameter $\beta = N_P kT/(B^2/8\pi)$, as well as angle $\gamma$. The thermal pressure and the parameter $\beta$ in this study were calculated without taking into account the contribution of the electron and helium components, since the meas-

urements of the parameters of these components may contain a large uncertainty. In a similar way, $N_pkT$ and $\beta$ were calculated, for example, in the OMNI database (https://omniweb.gsfc.nasa.gov/ftpbrowser/bow_derivation.html, accessed on 1 February 2022).

The angle $\gamma$ is an analog of the angle used in the paper [31] to determine the moment of the minimum distance between the spacecraft trajectory and the axis of the magnetic flux rope (MFR). At the minimum distance between the spacecraft and the MFR axis, the satellite moves tangentially to the local magnetic field of the flux rope, and the value of this angle is either 0° or 180° (see Figure 1). In this research, we calculated $\gamma$ as the angle between the magnetic field $\vec{\mathbf{B}}$ and the relative spacecraft velocity $\vec{\mathbf{V}}$. The velocity of the spacecraft, whose measurements are the main sources of helium data in the OMNI database (for example, WIND, ACE), is low compared to the SW bulk velocity. The solar wind propagation is close to radial [32]; therefore, the bulk velocity on average is directed along the Sun–Earth line, i.e., along the X-axis of the GSE coordinate system. Then, the satellite's relative velocity $\vec{\mathbf{V}}$ is directed along the same X-axis in the coordinate system associated with the solar wind flow. Thus, $\gamma$ is close to the angle between the X-axis and the magnetic field vector $\vec{\mathbf{B}}$ and characterizes the position of the satellite inside the IC-ME—the closer the $\gamma$ is to 90°, the farther from the ICME axis the spacecraft is located. Meanwhile, the value of $\gamma$ can be either in the range 0°–90°, or in the range 90°–180°. Since the relationship of the angle with the distance between the ICME axis and the spacecraft in these ranges is symmetric relative to $\gamma = 90°$, the value of the angle adjacent to $\gamma$ was taken into account in the subsequent analysis if $\gamma$ value was between 90° and 180°.

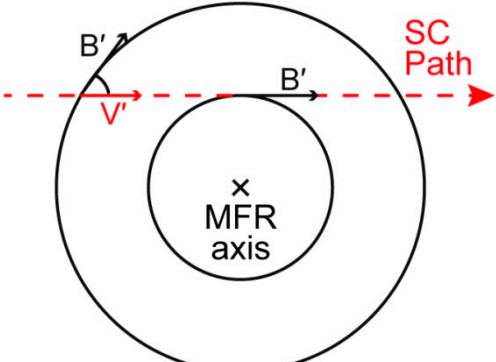

**Figure 1.** The schematic view of the magnetic flux rope (MFR) cross section (adapted from the paper by Zhang et al. [31]). The relative arrangement of components of the local magnetic field $\vec{\mathbf{B}}'$ and spacecraft velocity $\vec{\mathbf{V}'}$ perpendicular to the MFR axis orientation is shown.

For the analysis, the data on MCs and Ejecta were divided into two periods: 1976–1996 and 1997–2019, which correspond to different epochs of solar activity. The analysis includes the calculation of standard parameters: average values, standard deviations, statistical errors (standard deviations divided by the square root of the number of points), and medians of the helium abundance $N_\alpha/N_p$ for each case. In further analysis, we approximate the dependences of the helium abundance on different parameters either by power-law or linear functions. To assess the quality of the approximation, the method described in [33] was used, which was also previously applied in our papers [22,28]. Despite the low correlation between the parameters (see the next section), due to the large number of measurement points, the probability of accuracy of each approximation exceeds 95%, and we consider them quite reliable.

### 3. Results

The number of selected measurement points for both ICME types and during two periods of solar activity, as well as several statistical characteristics, is presented in Table 1. Due to the large number of points and small statistical error, it can be concluded with high statistical significance that (1) the helium abundance is higher in MC than in Ejecta for each epoch of solar activity, and (2) in the epoch of low solar activity, the helium abundance is much lower than in the era of high activity.

**Table 1.** Number of measurement points N, average values ⟨⟩, standard deviations δ, statistical errors SE, and medians M of the helium abundance $N_\alpha/N_P$ in each interplanetary coronal mass ejections (ICME) type (magnetic cloud (MC) and Ejecta) and period.

| ICME Type | MC | | Ejecta | |
|---|---|---|---|---|
| Period | 1976–1996 | 1997–2019 | 1976–1996 | 1997–2019 |
| N | 605 | 2157 | 7365 | 12,210 |
| $\langle N_\alpha/N_P \rangle \pm \delta$ (SE), % | 6.3 ± 5.2 (0.21) | 5.2 ± 3.6 (0.08) | 4.5 ± 3.3 (0.04) | 3.0 ± 2.2 (0.02) |
| M ($N_\alpha/N_P$), % | 5.1 | 4.4 | 3.9 | 2.5 |

The main results obtained are presented in this section in two figures. Figure 2 shows the relationship between the helium abundance $N_\alpha/N_P$ and the investigated parameters for MC: parameter β, IMF magnitude B, thermal pressure $N_P kT$ (panels a–c), protons density $N_P$, protons temperature T and angle γ (panels d–f). The dependences are shown in two rows of panels: the top row corresponds to data for the period 1976–1996 (SCs 21–22), the bottom row–for period the 1997–2019 (SCs 23–24). Each point on the panels represents a 1-h measurement point selected for analysis from the OMNI database (see the previous section).

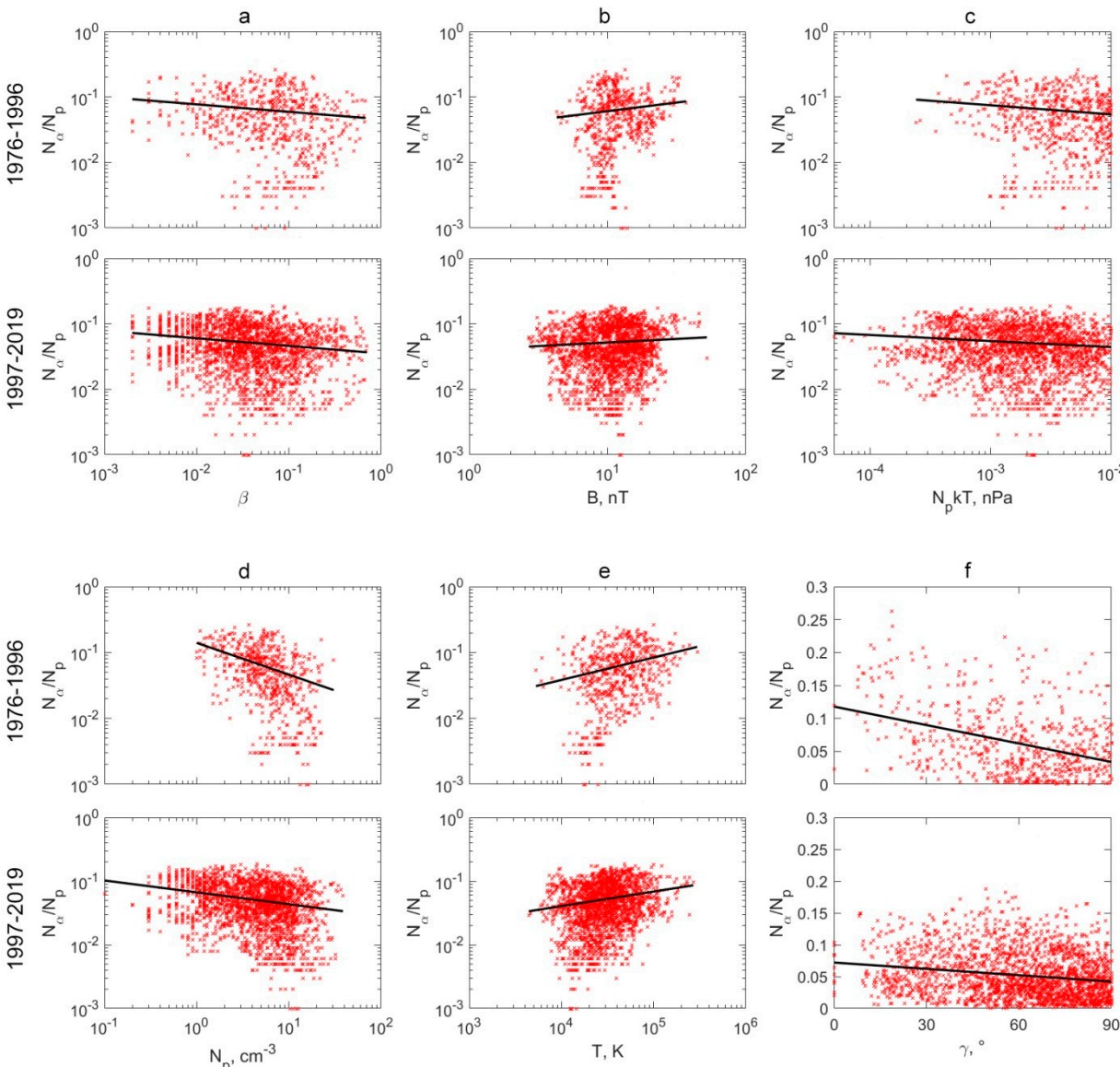

**Figure 2.** Dependences of the helium abundance N$\alpha$/Np on parameter $\beta$, interplanetary magnetic field (IMF) magnitude B, thermal pressure NpkT, proton density Np and temperature T, and angle $\gamma$ in MC in epochs of high (1976–1996) and low (1997–2019) solar activity. (**a**) parameter $\beta$; (**b**) IMF magnitude B; (**c**) thermal pressure NpkT; (**d**) protons density Np; (**e**) protons temperature T; (**f**) angle $\gamma$.

　　Black straight lines are approximations of the dependences. Similar to the paper [28], a logarithmic scale is used in panels a–e. This allows us to display a wide range of variables on the X-axis (for example, the proton temperature). The dependences represented on these panels were approximated by power functions, which gave straight lines on the corresponding panels. Since the angle $\gamma$ takes values from 0° to 90°, the dependence of N$\alpha$/N$_P$ on $\gamma$ is shown on a linear scale for greater clarity. Approximation of this dependence was carried out using a polynomial of the first degree.

　　Figure 2 shows that the number of points is noticeably different in the two periods. The number of measurements inside MC in the period 1997–2019 is 3.5 times more than in the period 1976–1996 (see Table 1). A similar situation is observed in Ejecta, but with a smaller difference of about 1.6 times. The main reason for this difference is that the time of simultaneous observations of plasma and magnetic field parameters in the OMNI database was often less than 60% of the annual time in the period 1976–1995, and has increased to more than 90% since 1996 [22].

The general trends in the dependences of the helium abundance on the considered parameters in MC are the same in both epochs of solar activity. As a rule, the approximation lines for the period 1976–1996 have a steeper slope; however, the slope for the $N_\alpha/N_P$ dependence on $\beta$ and $N_PkT$ is almost the same in the two periods. Figure 2b,c show how the helium abundance depends on the magnetic and thermal pressures. The $N_\alpha/N_P$ value increases with increasing magnetic field magnitude B. As a consequence, the $N_\alpha/N_P$ dependence on the magnetic pressure $B^2/2\pi$ is increasing. Meanwhile, the helium abundance decreases with increasing thermal pressure $N_PkT$, which ultimately leads to a decreasing dependence of $N_\alpha/N_P$ on parameter $\beta$, i.e., the ratio of $N_PkT$ to $B^2/2\pi$. The dependences of the relative helium abundance on the thermal pressure components have the following form: decreasing for the proton density $N_P$ (according to the definition of $N_\alpha/N_P$) and increasing for the temperature T. Figure 2e shows a clear dependence of the helium abundance on the angle $\gamma$ in MC. The $N_\alpha/N_P$ value increases as $\gamma$ decreases, reaching a maximum at $\gamma = 0°$. For this pair of parameters, the largest difference between the approximations in the two periods is observed: the slope of the approximation line in the period 1976–1996 is ~2.8 times more than in the period 1997–2019.

There were no significant differences in the dependences of the helium abundance on the same parameters for Ejecta in comparison with the dependences for MC. In addition, too many points make the figures less clear. In view of this, we do not present a figure for Ejecta events similar to Figure 2 in the paper.

For a visual comparison of the data between MC and Ejecta and epochs of low and high solar activity, the approximations of dependences for all four cases are shown in Figure 3. The color of the line indicates the type of ICME: red for MC and blue for Ejecta. Thin lines correspond to the period of SCs 21–22, thick lines correspond to cycles 23–24. The explicit form of the approximation function for each of the dependences is given in Table 2. It also shows the Pearson correlation coefficients r between the helium abundance and the corresponding parameters.

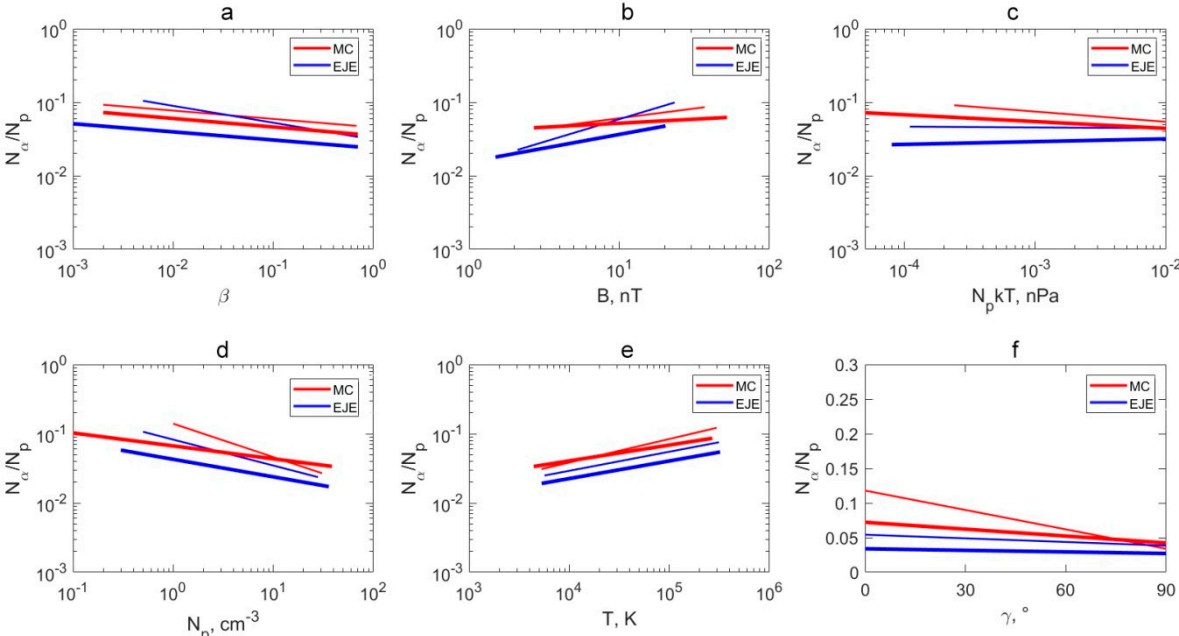

**Figure 3.** Approximations of dependences of the helium abundance $N\alpha/Np$ on plasma and IMF parameters (similar to Figure 2) for MC (red lines) and Ejecta (blue lines) in the period 1976–1996 (thin lines) and the period 1997–2019 (thick lines). (**a**) parameter $\beta$; (**b**) IMF magnitude B; (**c**) thermal pressure NpkT; (**d**) protons density Np; (**e**) protons temperature T; (**f**) angle $\gamma$.

**Table 2.** The Pearson correlation coefficients r and approximations for each of the dependences shown in Figure 3.

| ICME Type | | MC | | Ejecta | |
|---|---|---|---|---|---|
| Period | | 1976–1996 | 1997–2019 | 1976–1996 | 1997–2019 |
| $N_\alpha/N_P$ on $\beta$ | r | −0.16 | −0.11 | −0.23 | −0.08 |
| | Appr. | $\ln y = -0.11 \times \ln x - 3.09$ | $\ln y = -0.12 \times \ln x - 3.34$ | $\ln y = -0.23 \times \ln x - 3.47$ | $\ln y = -0.11 \times \ln x - 3.74$ |
| $N_\alpha/N_P$ on B | r | 0.13 | 0.10 | 0.29 | 0.20 |
| | Appr. | $\ln y = 0.26 \times \ln x - 3.41$ | $\ln y = 0.11 \times \ln x - 3.21$ | $\ln y = 0.62 \times \ln x - 4.25$ | $\ln y = 0.37 \times \ln x - 4.17$ |
| $N_\alpha/N_P$ on $N_P kT$ | r | −0.12 | −0.15 | 0.04 | 0.07 |
| | Appr. | $\ln y = -0.14 \times \ln x - 3.56$ | $\ln y = -0.09 \times \ln x - 3.55$ | $\ln y = -0.01 \times \ln x - 3.17$ | $\ln y = 0.04 \times \ln x - 3.28$ |
| $N_\alpha/N_P$ on $N_P$ | r | −0.39 | −0.24 | −0.28 | −0.15 |
| | Appr. | $\ln y = -0.48 \times \ln x - 1.97$ | $\ln y = -0.19 \times \ln x - 2.71$ | $\ln y = -0.37 \times \ln x - 2.5$ | $\ln y = -0.25 \times \ln x - 3.15$ |
| $N_\alpha/N_P$ on T | r | 0.26 | 0.19 | 0.22 | 0.22 |
| | Appr. | $\ln y = 0.34 \times \ln x - 6.38$ | $\ln y = 0.23 \times \ln x - 5.32$ | $\ln y = 0.28 \times \ln x - 6.07$ | $\ln y = 0.25 \times \ln x - 6.12$ |
| $N_\alpha/N_P$ on $\gamma$ | r | −0.38 | −0.19 | −0.12 | −0.08 |
| | Appr. | $y = -9.34 \times 10^{-4} \times x + 0.12$ | $y = -3.32 \times 10^{-4} \times x + 0.07$ | $y = -1.76 \times 10^{-4} \times x + 0.05$ | $y = -0.76 \times 10^{-4} \times x + 0.03$ |

Several features can be distinguished in Figure 3. Firstly, the general trends of the dependences of the helium abundance on the other parameters are the same in the two periods not only for MC, but also for Ejecta. The nature of all dependences (decreasing or increasing), except for thermal pressure in Ejecta, is similar to dependences in MC. The helium abundance is almost independent of $N_P kT$ in Ejecta (the exponent modulus for both periods is < 0.05). Secondly, a higher slope of approximation of the $N_\alpha/N_P$ dependence on B in Ejecta is observed in the epoch of high solar activity, the same as in MC. Thirdly, the dependence of the relative helium abundance on the angle $\gamma$ in Ejecta is almost absent, while the approximation lines have a significant slope for MC, and the slope is higher in the epoch of high solar activity. The minimum value of $N_\alpha/N_P$, approximately 3–4%, is reached at $\gamma = 90°$ for all four cases.

### 4. Discussion

The dependences of the relative helium abundance $N_\alpha/N_P$ on the parameters $\beta$, B, $N_P kT$, $N_P$ and T, obtained as a result of the analysis, are generally consistent with the results of the previous study [28], where similar dependences were considered for the full period 1976–2019.

When comparing the approximations of the dependences in MC and Ejecta separately in each period, Figure 3 shows that the lines for MC are almost always located higher. This indicates higher average helium abundance in MCs compared to Ejecta, which is also reflected in Table 1 and was shown, for example, in the papers [19,22]. As expected, this ratio did not change during the transition from high to low solar activity. In our paper [19], it was shown that this ratio of the helium abundance between MC and Ejecta is fulfilled not only on the scales of SC phases and full ICME intervals [25], but also on hourly scales during the analysis of time profiles by the double superposed epoch method. This conclusion, in particular, means that the criterion $N_\alpha/N_P > 8\%$, used in the identification of magnetic clouds in some early research [34–36], ceases to be correct in the epoch of low solar activity.

The dependences of the helium abundance $N_\alpha/N_P$ on the interplanetary magnetic field magnitude B, the proton density $N_P$, and the angle $\gamma$ became weaker in the epoch of low solar activity. An increasing dependence of $N_\alpha/N_P$ on B is observed in all four cases considered. Since the $N_\alpha/N_P$ dependence on $N_P kT$ is decreasing for MC and almost absent for Ejecta, an anticorrelation between the helium abundance and parameter $\beta$, which is

the ratio of thermal and magnetic pressures, is observed for MC and Ejecta in both periods.

The approximation lines corresponding to the dependences for the period 1996–2019 lie mainly below the approximations for the period 1976–1996 for both types of ICMEs. This fact indicates that the helium abundance decreased in SCs 23–24 compared to SCs 21–22 in these SW phenomena. Table 1 shows that the average $N_\alpha/N_P$ value decreased by ~15% in MC and ~35% in Ejecta. These results indicate a significant drop in the relative helium abundance inside ICMEs during the minimum of solar activity between SCs 22 and 23, which is consistent with the results of the papers [19,25], and specifies a general decrease in the activity of the solar atmosphere due to the formation of minor ion components of solar wind.

As noted above, with a general drop in the helium abundance in SCs 23–24, some of the dependences of $N_\alpha/N_P$ on the solar wind parameters became somewhat weaker. However, the general trends of the dependences have not changed, which was shown for the first time and is the most important result of this research. Based on this result, it can be assumed that the kinematics of CME formation on the Sun and the ICME dynamics in interplanetary space can change insignificantly [37].

The trajectory of a spacecraft inside the coronal mass ejection largely determines which of the two subtypes of ICMEs a particular structure will be assigned to [23]. When crossing the Ejecta, it passes through the peripheral regions of the ICME, and in the case of the MC, it lies closer to the axis of the structure. This means that the same value of the angle $\gamma$ for MC and Ejecta corresponds to a different distance between the spacecraft and the ICME axis. However, for each case when the spacecraft crosses the MFR, the distance from the satellite to the MFR axis must be the minimum for a particular trajectory [31]. In our analysis, we summarize the measurements for many trajectories without knowing the actual configuration of the satellite's trajectory relative to the flux rope. Nevertheless, under the assumptions and simplifications made, we found $N_\alpha/N_P$ dependences on the angle $\gamma$ with the same sign of the slope. In Ejecta, the dependences are very weak, while in MC they are stronger, and the strongest dependence is observed in the epoch of high solar activity. The helium abundance anticorrelates with the angle $\gamma$ in MC and reaches its maximum values at $\gamma < 15°$ (Figure 3f), when, according to the model, the distance to the ICME axis is minimal. The revealed regularity confirms the hypothesis discussed in the papers [22,29] about the existence of an electric current enriched with helium inside ICMEs. Low $N_\alpha/N_P$ dependences on the angle $\gamma$ in Ejecta may be due, in part, to the violation of the MFR configuration assumption in the peripheral region of ICME.

## 5. Conclusions

In this study, based on 1-h data from the OMNI database, we studied the behavior of the relative helium abundance inside interplanetary coronal mass ejections ICMEs over the last four SCs. In particular, the dependences of the helium abundance $N_\alpha/N_P$ on the plasma and interplanetary magnetic field parameters in MCs and Ejecta were investigated during the epoch of high solar activity in SCs 21–22 (1976–1996) and the epoch of low activity in SCs 23–24 (1997–2019). Comparison of dependences showed the following results:

1. A significant decrease in the relative helium abundance is observed inside ICMEs in SCs 23–24 compared to SCs 21–22. Such a decrease in the helium abundance was previously shown in the paper [25], which studied the behavior of plasma and magnetic field parameters in different types of solar wind streams and at different phases over SCs 21–24, and in the paper [19] on the analysis of time profiles of parameters. This observational fact is one of the consequences of the decline in solar activity at the end of the 20th century;

2. The anticorrelation between the helium abundance and the plasma parameter β is observed in two types of ICMEs (MCs and Ejecta) in both periods. It is related to the

increasing dependence of the helium abundance on the interplanetary magnetic field magnitude, which becomes weaker in the epoch of low solar activity compared to the epoch of high activity. The obtained dependences are in good agreement with the results of the previous study [28], performed without taking into account the influence of changes in solar activity, and complement its conclusions;

3. It is shown that the helium abundance in an ICME changes with distance from the coronal mass ejection axis: $N_\alpha/N_P$ takes the highest values near the axis, and it almost does not change in the peripheral regions. The results obtained by a method similar to that used in [31] confirm the hypothesis of the existence of a helium-enriched current at the center of the ICME, which was suggested in the paper [22];

4. The general nature of the dependences of the helium abundance on the solar wind plasma and interplanetary magnetic field parameters (for example, the parameter $\beta$ and the magnetic field magnitude), observed in the epoch of high activity, was generally preserved in the epoch of low activity, despite the fact that some dependences became weaker compared to SCs 21–22.

Thus, the solar activity decrease in the SCs 23 and 24 led to a significant change in the heliosphere [25] and a decrease in the relative helium abundance in ICMEs. Experiments have shown that the minor ions abundance in the electric current inside the solar wind plasma and the Earth or Mars magnetosphere exceeds the abundance in the background plasma. There was a suggestion about the fundamental plasma law consisting of the minor ions abundance increasing in current sheets [11,14,38–43]. The results obtained in this study show that the decrease in the relative helium abundance in the ICME current sheet occurs simultaneously with its drop in the background plasma, in quasi-stationary fast and slow solar wind streams. This observation allows us to suggest that the helium abundance value in the current sheet depends on the helium abundance in the background plasma of the solar atmosphere. Generally, changes in the properties and structure of the heliosphere can have a noticeable effect on large-scale solar wind phenomena. This should be taken into account when studying space weather, and, in particular, when predicting the motion and properties of ICMEs during their interaction with the Earth's magnetosphere.

**Author Contributions:** Conceptualization, A.A.K.; data curation, I.G.L. and L.S.R.; methodology, Y.I.Y.; software, A.A.K. and I.G.L.; supervision, Y.I.Y.; validation, M.O.R.; visualization, A.A.K.; writing—original draft, A.A.K.; writing—review and editing, A.A.K., Y.I.Y. and M.O.R. All authors have read and agreed to the published version of the manuscript.

**Funding:** The study was supported by the Russian Science Foundation, grant 22-12-00227.

**Acknowledgments:** The authors thank the creators of databases https://spdf.gsfc.nasa.gov/pub/data/omni/low_res_omni (accessed on 1 February 2022) and http://www.iki.rssi.ru/pub/omni (accessed on 19 April 2022) for the possibility to use them in the research.

**Conflicts of interest:** The authors declare no conflict of interest.

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
