# Peer review of "Helium Abundance Decrease in ICMEs in 23–24 Solar Cycles"

_universe, doi:10.3390/universe8110557_

Round 1

Reviewer 1 Report

The manuscript 'Helium Abundance Decrease in ICMEs in 23-24 Solar Cycles' presents a decrease in Helium detected during periods in which ICMEs were detected, as well the relationship between the magnitude of this decrease and various other parameters. The results are presented clearly and make sense, though I do question the novelty of the work given nearly every main conclusion of the paper includes a statement along the lines of 'matching what was shown in a prior paper'. I think the authors could do more to make the physical significance of what they are presenting more obvious.

The data certainly support the notion that there are fewer alphas detected in ICMEs in the more recent solar cycles. What is never stated is the physical significance of this decrease and the impact it might have on CME physics. In the discussion between lines 216-218 it is stated that the decrease in alphas changes the criterion for magnetic cloud identification. This could seriously impact ICME identification attempts and could be a significant impact of this study. But does the change in Helium within the events carry any deeper physical meaning?

I think the methodology is sound and the paper could be published more or less as it is, but would greatly benefit from a far more detailed discussion of the actual physical implications of the alpha decrease in the last few solar cycles. 

Detailed Comments Below:

Abstract: The final line, and then the text itself, mention this helium-enriched electric current but only really in passing. If the confirmation of this current is meant to be the most significant result of this paper, it should be explained in more depth here than with just a reference to another paper.

Line 58: Locate -> Located

Line 60-61: When periods of high activity and low activity are discussed, it's usually in reference to solar minimum/maximum. Here it means the overall strength of the various cycles. I think it would be helpful to include a solar activity proxy like SSN or S10 flux over the 4 solar cycles to demonstrate the weakened overall activity and show exactly what you mean when discussing the different activity levels.

Line 46-61: I think this paragraph summarizes one of my concerns with this paper. The decrease in alpha's of 30% has been established in a prior paper. The decrease in plasma and magnetic parameters has been established in another paper. Isn't this paper just taking those published results and combining them in a fairly unsurprising way?

Line 64 and 67: insert 'an' before discussing a singular ICME

Line 83: remove S after ICME

Line 101: It took a second, but I understood the angle you're referring to after I sketched it out. A diagram of this angle in a CME reference frame would be helpful

Line 117: include -> includes

Line 210: add 'are' between MC and almost

Line 289-292: How specifically would this change in composition impact ICME kinematics? This is an incredibly vague motivation for this study and I think including some specificity here would address a lot of my concerns about the significance of the conclusions.

Reviewer 2 Report

This paper reports on the results of a statistical analysis about ICME helium abundance relative to that of protons by comparing between low and high solar activity epochs. The results appear to make sense from intuitive ground and consistent with previous reports. One interesting new finding is the dependence of the relative helium abundance on the angle between the sun-earth line and local magnetic field direction (the gamma parameter). From this, the authors argue that the relative helium abundance is maximum at the minimum distance between spacecraft and ICME flux rope axis. Since this is the main new finding, one main question below needs to be explained further before I can finally evaluate this manuscript.

In page 3, the authors state the following and more statements follow there:

“At the minimum distance between the spacecraft and the MFR axis, the satellite moves tangentially to the local magnetic field of the flux rope, and the value of this angle is either 0° or 180°.”

Frankly this was confusing to me. My understating is that MC flux ropes usually have helical field lines and the field line at the axis is more like toroidal and those near the edge may have more poloidal field lines. In addition, the orientation of flux rope axis relative to the X(GSE) axis can be arbitrary. Considering all these, the relative angle between X(GSE) axis and the local B inside MCs can be arbitrary irrelevant to the distance of spacecraft to the flux rope axis. It would be useful if this issue is more clearly demonstrated in the manuscript.

If the issue above is clarified, since the helium abundance dependence on the angle is the main result, I suggest it to be more emphasized in abstract by rephrasing appropriately, for example, something like “the enhancement in relative helium abundance is clearer for MCs than Ejecta for both solar activity epochs and clearest for MCs in high solar activity”.

Minor issues:

Line 27: “…varied variations of…” à “…various variations of…” may read better.

Line 29: “small scales (less than 105 km),” à please be more specific what this 105 km threshold means, what it refers to, or where it comes from, say if it is ever related to particle gyroradius, inertial length, minimum size of ICMEs, or whatever. Incidentally there are far flux ropes at small scales (by 100s times) than ICMEs for your reference.
